# Strategies to Develop a Suitable Formulation for Inflammatory Skin Disease Treatment

**DOI:** 10.3390/ijms22116078

**Published:** 2021-06-04

**Authors:** Jiun-Wen Guo, Shiou-Hwa Jee

**Affiliations:** 1Department of Medical Research, Cathay General Hospital, Taipei 10630, Taiwan; 2Program in Pharmaceutical Biotechnology, College of Medicine, Fu Jen Catholic University, New Taipei City 24205, Taiwan; 3Department of Dermatology, Cathay General Hospital, Taipei 10630, Taiwan; shiouhwa@ntu.edu.tw; 4Department of Dermatology, College of Medicine, National Taiwan University, Taipei 10617, Taiwan

**Keywords:** skin barrier, inflammatory skin, topical treatment, stratum corneum, formulation

## Abstract

Skin barrier functions, environmental insults, and genetic backgrounds are intricately linked and form the basis of common inflammatory skin disorders, such as atopic dermatitis, psoriasis, and seborrheic dermatitis, which may seriously affect one’s quality of life. Topical therapy is usually the first line of management. It is believed that successful topical treatment requires pharmaceutical formulation from a sufficient dosage to exert therapeutic effects by penetrating the stratum corneum and then diffusing to the target area. However, many factors can affect this process including the physicochemical properties of the active compound, the composition of the formulation base, and the limitations and conditions of the skin barrier, especially in inflammatory skin. This article briefly reviews the available data on these issues and provides opinions on strategies to develop a suitable formulation for inflammatory skin disease treatment.

## 1. Introduction

The skin, which is the largest organ, comprising of the epidermis, dermis, and hypodermis layers, covers the body’s entire external surface. Stratum corneum, the epidermis’ most superficial layer, serves as a protective barrier against the external environment and protects inner cells from dehydration, infection, and physical, chemical, and mechanical insults [1,2]. The stratum corneum is made up of corneocytes, which are anucleated keratinocytes within a lipid-rich extracellular matrix. Peter Elias described this structure as the “bricks and mortar” model, whereby the bricks are the corneocytes and the mortar refers to the surrounding lipids [3]. Skin barrier integrity can be influenced by various factors such as differentiation, proliferation, and adhesion of epidermal cells and skin lipids. Dysfunction of the skin barrier can cause skin disorders, for example, atopic dermatitis, psoriasis, and eczema. On the contrary, skin disorders can also impair the skin barrier [4,5]. Specifically, the signaling molecules released from the injured stratum corneum initiate a cytokine cascade, triggering an inflammatory response, which then contributes to the pathogenesis of a variety of dermatoses [6,7].

On the other hand, the stratum corneum also works as the first limiting barrier for drugs to be transported into the skin [8]. The skin barrier can resist the penetration of many molecules. The 500 Dalton rule proposes that for skin absorption, a compound must be under 500 Dalton and larger molecules cannot pass the corneal layer [9]. Therefore, different strategies have been developed to overcome this issue and achieve successful topical drug delivery [8].

However, for inflammatory skin disorder treatment, topical delivery strategies must be extremely careful to prevent inducing a further inflammatory reaction whilst achieving effective drug delivery to the target site. Thus, this review focuses on and briefly reviews the available data on these issues whilst providing opinions on strategies to develop a suitable formulation for inflammatory skin diseases treatment.

## 2. Inflammatory Skin Diseases

### 2.1. Inflammatory Skin Diseases, Skin Barrier, and T-Cells Immune System

Skin barrier, innate immunity and acquired immunity are the three major components of the skin’s host defense system. The dysfunction of any of these components may lead to inflammatory skin disorders, unless there is a general response to specific infectious pathogens or internal/external damage [10]. As mentioned in the introduction, the stratum corneum serves as a protective barrier against the external environment and protects inner cells from physical, chemical, and mechanical insults [1,2]. Dysfunction of the skin barrier may lead to a skin inflammation response whereby the signaling molecules are released from the injured stratum corneum, initiating a cytokine cascade and triggering an inflammatory response, which then contributes to the pathogenesis of a variety of dermatoses [6,7].

The non-specific innate responses immediately work to defend against anything foreign. Physical barriers such as the tight junctions between the stratum corneum, white blood cells such as neutrophils, basophils, eosinophils, monocytes, macrophages, reticuloendothelial system, and natural killer cells, as well as membrane-bound receptors and intracellular proteins, are all parts of innate immunity [11]. The adaptive-acquired immune responses, including the specific lymphocytes and their products, such as immunoglobulins and cytokines, generate a specific response triggered by the particular pathogens, immunogens, or antigens. Additionally, the acquired immunity can identify the pathogen quickly and creates a faster and more intensive response upon re-contact [11]. An effective skin immune response requires antigen-specific T cells located in the damaged or infected skin area. Memory T cells are involved in the quick, intensive, and long-lasting response against re-contact immune challenges. Among the memory T cells, tissue-resident memory T cells remain in the skin for long periods of time and involve lasting protective immunity as well as the regulation of several inflammatory skin diseases [12].

### 2.2. T Cells-Mediated Inflammatory Skin Diseases

Based on the involved T cell subtypes, T-cell-mediated diseases can be classified into T helper 1 (Th1) cell-dominated such as vitiligo, T helper 2 (Th2) cell-dominated such as atopic dermatitis, and Th1/T helper 17 (Th17) cell-involved responses such as psoriasis, and regulatory T (Treg) cell-based responses such as melanoma [13].

#### 2.2.1. Vitiligo

Vitiligo is an acquired immune-dysregulation skin disorder characterized by the loss and degradation of functional epidermal melanocytes. Confetti, trichrome, inflammatory lesions, and koebnerization are the most common clinical futures. The estimated prevalence of vitiligo is in about 1% of the global population. Vitiligo may cause a harmful psychological stamp, especially in dark-skinned people. It can affect the skin on any part of the body but commonly occurs on the face, neck, and hands and in skin creases. It can also affect the hair and the inside of the mouth [14,15]. A T helper 1 cell 1/cytotoxic T (Tc1) cell 1-dominated immune response is well demonstrated in vitiligo. Recent studies reported that vitiligo patients had elevated numbers of circulating Th1/17 cells and Tc1/17 cells [16,17]. A previous study reported that the vitiligo-involved site significantly delayed the barrier recovery rate compared with the uninvolved sites after tape stripping [18]. This result may imply an unbalanced barrier homeostasis/impaired barrier function in vitiligo. The treatment strategy for vitiligo is to maintain the disease phase and prevent relapse depending on the clinical classification/characteristics and repigmentation of the involved disease area [19]. Phototherapy and topical agents such as corticosteroids and calcineurin inhibitors are used as the main treatment methods [20].

#### 2.2.2. Atopic Dermatitis

Atopic dermatitis is one of the most common chronic relapsing inflammatory skin diseases. Approximately 15% to 20% of children and 1% to 3% of adults worldwide suffer from this disease. The main factors affecting the development of atopic dermatitis are complex and multifaceted, including genetic defects, an impaired skin barrier, abnormal immune system regulation, IgE-mediated hypersensitivity and environmental factors. The injured stratum corneum may be the initial event that leads to further skin inflammation and allergic responses [21,22]. Increased trans-epidermal water loss value, changes to the skin surface pH, and skin dehydration may lead to severe atopic dermatitis because of the loss of function or mutations in filaggrin. Th2 cells release cytokines including interleukin 2 (IL-2), IL-4, as well as IL-17 and IL-22 released from Th17 and Th22 cells, which may all contribute to skin barrier disruption and the development of atopic dermatitis. The Th2 cells mediated immune responses may also boost the IgE-mediated hypersensitivity response and facilitate atopic dermatitis development [21,22]. The itching sensation from sensory nerves may trigger repeated scratching behavior, which may lead to a skin barrier dysfunction loop, which then affects the patient’s quality of life [21,22,23]. Topical corticosteroids, in conjunction with calcineurin inhibitors such as pimecrolimus and tacrolimus, are the first-line treatment for atopic dermatitis. Antibiotic treatment is effective in treating or preventing secondary skin infections, whilst ultraviolet phototherapy is another safe and effective treatment option for moderate to severe atopic dermatitis. However, oral antihistamines are not recommended for atopic dermatitis treatment because of their limited pruritus symptom inhibition. Maintenance therapy such as daily topical emollients or bathing with soap-free cleansers are also recommended in conjunction with conventional treatment [23].

#### 2.2.3. Psoriasis

Psoriasis is an immune-mediated genetic disease that causes red, flaky, and crusty skin patches covered with silvery scales. It normally appears on elbows, knees, the scalp, and the lower back. Psoriasis affects people of all ages and is most common in the age group of 15 to 30 in all countries [24]. Guttate, erythrodermic, pustular, inverse, and psoriasis vulgaris are the major clinical variants of psoriasis. Approximately 90% of patients represent psoriasis vulgaris. The severity can be divided into benign, moderate, and severe psoriasis. The most frequent histopathological findings in psoriasis are inflammatory cell infiltration, vascular dilatation, absence of the granular layer, regular elongation of rete ridge, elongation of the dermal papillae, and parakeratosis [24]. The possible pathogenesis of psoriasis demonstrates a complex interaction between the innate and adaptive immune systems. Specifically, T cells may cross-talk with dendritic cells, macrophages, and keratinocytes, mediated by their secreted proinflammatory cytokines such as tumor necrosis factor-alpha (TNF-α), IL-1 and IL-6 [24]. Genetic, immunological, and environmental factors are related to the pathogenesis of psoriasis. Various predisposing factors may cause psoriasis in genetically susceptible populations. It is suggested that psoriasis may be associated with other auto-inflammatory and auto-immune diseases [25]. However, the exact cause of psoriasis is not yet well understood [24].

About 75% of all psoriasis patients can be appropriately treated with topical glucocorticosteroids, vitamin D derivatives, or combinations of both [24,26]. However, the practicability, convenience, and adverse effects such as skin irritation may limit the use of topical drugs [24]. Phototherapy and systemic treatments are other effective methods of psoriasis management. However, the cumulative toxicity potential of individual therapy limits the duration of treatment [27]. For example, phototherapy may increase the risk of cancer, whilst methotrexate may cause adverse reactions such as nausea and vomiting [28]. Furthermore, sometimes the treatment effect may decrease over time and therefore requires replacement with alternative therapies [29].

The recently developed and approved biological drugs targeting TNF-α, IL-23, and IL-17 provide another choice for psoriasis treatment. The inflammatory response induced by the IL-23/Th17 axis can be blocked by either directly or indirectly inhibiting IL-17-producing cells or their receptors. However, the use of biological drugs has dramatically changed the treatment and management of psoriasis [30]. At this present time, there is still no curative treatment for psoriasis.

#### 2.2.4. Melanoma

Melanoma is a serious type of skin cancer that develops from melanocytes. Melanoma typically occurs in the skin but can also occur, although rare, in the mouth, intestines, or eye with irregular sharpness and irregular color. It primarily occurs after DNA mutation, most often due to excess sun/ultraviolet light exposure to fair-skinned or light-haired areas [31]. Melanoma could be considered a highly immunogenic tumor. Innate immunity played a key role in the development, growth, and prognosis of melanoma. Compared with the normal un-lesion skin site, amounts of active macrophages, dendritic cells, mast cells, and natural killer cells were redistributed at the melanoma lesion site [32,33]. Langerhans cells, melanocytes, and Merkel cells are three main resident dendritic morphology cells in the skin. Langerhans cells are known to have the capacity to present antigens and are essential for initiation and maintenance of specific T-cell-mediate responses in the skin [32,34]. Langerhans cells are related to early transformed melanocytes. A previous study reported that the numbers of melanoma-associated Langerhans cells decreased as melanoma progressed [35]. The main function of melanocytes is to synthesize and provide melanin to keratinocytes. Sun-exposure-related melanoma has high melanocyte mutations such as NRAS, neurofibromin 1, KIT, and BRAF^nonV600E^ compared with common mutation BRAF^V600E^ in non-sun-exposed-related melanoma [36,37]. Merkel cells are regarded to play an essential role in sensory discernment. Ultraviolet light exposure might also lead Merkel cell mutations to progress as Merkel cell carcinoma [37]. Regulatory T cells are the critical mediators of immune suppression in the tumor microenvironment, although their role in suppressing immune surveillance during tumorigenesis currently has limited understanding [38]. Zochake et al. reported an impaired barrier function in non-melanoma skin cancer results from the alteration of stratum corneum lipid packing and profile, such as a decreased cholesterol content, an increased phospholipid amount, and the composition of ceramide subtypes [39]. However, to our best knowledge, there are few scientific reports about skin barrier function in melanoma.

### 2.3. T-Cell-Involved Inflammatory Skin Barrier

Naive T cells can be differentiated into several different types of effectors and regulatory T cells. Specific cytokines and transcription factors contribute to the differentiation and expansion of these effectors and regulatory T cell populations. Their differential activation plays an important role in determining whether the immune response contributes to either host protection or pathological inflammation [21,22,40].

Cytokines IL-12, IL-4, and transforming growth factor-β (TGF-β) have been shown to regulate Th1, Th2, and Treg cells development, whilst IL-6, IL-23, and TGF-β have been shown to promote Th17-cell differentiation, respectively. Unlike Th1, Th2 and Th17 cells, which may mediate harmful skin inflammation processes, Treg cell are involved in the down-regulation immune response. TNF-α, interferon-gamma (IFN-γ), IL-22 product form Th1 and Th22 cells, respectively, leading to abnormal keratinocyte proliferation. Th2 cells produce a panel of cytokines such as IL-4 and IL-13, which also contribute to abnormal keratinocyte proliferation. In response to IL-4 and IL-13, B cells produce high amounts of IgE. Cutaneous resident and infiltrated cells release Th2-related chemokines such as cc chemokine ligand 17 (CCL17), CCL22, and CCL26. Th2 cells also release IL-31, which may stimulate sensory nerves, triggering the itch sensation. The itching leads to mechanical scratching behavior, causing barrier disruption [21,22,40].

In the IL-23/Th17 axis, the activated Th17 cells product cytokines IL-17A and IL-17F stimulate keratinocytes. The stimulated keratinocytes lead to abnormal differentiation and proliferation as well as an elevated production of proinflammatory cytokines such as IL-1 and TNF-α, antimicrobial peptides, and chemokines such as CCL20. The above keratinocyte-derived factors, in turn, stimulate further recruitment of Th17 cells and dendritic cells, which then establish a harmful inflammatory feedback loop [12,14,21,22,40]. The immune responses and abnormal keratinocyte differentiation and proliferation lead to a dysfunctional skin barrier and skin dehydration. The impaired skin barrier and dysregulated or misdirected immune response may result in chronic inflammatory skin.

To provide a global view of the relationship between T cells and the inflammatory skin barrier, Figure 1 illustrates the T cells involved in immune responses and the mechanisms of barrier dysfunction.

### 2.4. Topical Treatment and Inflammatory Skin Diseases

Treatment goals of inflammatory skin diseases are mainly symptom control and improving quality of life. The treatment options for inflammatory skin diseases include corticosteroids; vitamin D3 analogues; disease-modifying anti-rheumatic drugs, such as methotrexate and cyclosporine; and newly developed biological-targeted drugs targeting on the IL-23/Th-17 axis, such as brodalumab and risankizumab [41,42]. Both expensive biologics and systemic treatment may cause serious side effects. For example, methotrexate and retinoids are teratogenic and should never be used during pregnancy [40]. Well-known toxicity signs of methotrexate include bone marrow suppression, hepatic fibrosis, cirrhosis, and both oral and gastrointestinal ulceration [43,44]. Topical application of methotrexate may help avoid the systemic adverse effects of oral methotrexate for psoriasis treatment [45,46]. Therefore, topical treatment for local inflammatory skin diseases is considered safer to use. As such, a suitable topical delivery formulation is necessary for appropriate selection and development to improve therapeutic effects and treatment adherence, as well as to reduce side effects.

## 3. Skin Microbiota

With the advances of computing and high-throughput bacterial 16S rRNA genes sequencing technology, scientists in microbiology and dermatology can now analyze, identify and characterize different microbiota compositions in depth. The new subject of the role/functionality of skin microbiota in cutaneous disorders and the cross-talk network with immune responses and skin barrier functions can now discover significantly more than it could last decade. Bacteria, fungi, viruses, and mites are major components of the skin microbiome. Different, healthy skin sites have different diverse microbe communities. For example, the Corynebacterium and Staphylococcus species predominate in moist microenvironments, while the Propionibacterium species is most present in the sebaceous glands. Allergic or inflammatory status may arise when changes in steady microbiome occur [47,48,49].

The term “commensal-epidermal homeostasis” means the homeostatic interactions between normal skin commensal microbiota and the epidermis physical barrier. The epidermis physical barrier prevents the invasion of external pathogens and provides habitation to the commensals [48,50]. For example, the natural moisturizing factors and metabolites from skin commensal microbiota contribute to acidic skin pH. Therefore, failure to maintain acidic skin pH may alert the diversity of skin microbiota [48]. Therefore, the unbalance of commensal-epidermal homeostasis may affect the cross-talk of epidermal commensal-specific Th1 and Th17 cells and the secretion of antimicrobial peptides from keratinocytes and then lead to the barrier disruption [48,49]. External stimulus and pathogens may then further impairer the skin barrier and result in inflammatory skin diseases.

Decreased commensals abundance may contribute to the progression and exacerbation of inflammatory skin diseases, such as atopic dermatitis [48]. However, studies reported that the richness and diversity of skin microbiota could be reversed by topical emollients [51,52,53,54]. Thus, topical treatment to restore the skin barrier function is the primary prevention for developing inflammatory skin disease; however, selecting suitable topical formulation ingredients to maintain commensal-epidermal homeostasis is necessary.

## 4. Topical Formulation Development Strategies

### 4.1. Physicochemical Properties of the Active Compound

It has been widely accepted that substances with a molecular weight < 500 Dalton and a log P > 1 are able to penetrate through the lipid-rich stratum corneum and enter the viable epidermis [9,55]. In contrast, substances with a molecular weight > 500 Dalton and a log P < 1 are too hydrophilic to penetrate the barrier of the stratum corneum [55,56]. However, substances need the “carrier/delivery vehicles” to be administrated/applied on the skin’s surface and to diffuse/penetrate the target area of the epidermis/dermis. Solvents such as ethanol, propylene glycol and both dimethyl sulfoxide and ointments are the most common topical vehicles for lipophilic substances, yet solvents are limited as they are too fluid to be adsorbed on the skin surface. Experimental data demonstrates that solvents such as polyethylene glycol, propylene glycol, alcohols, dimethyl sulfoxide, and dimethyl acetamide may cause dehydration of the horny layer of the epidermis [57,58]. The therapeutic approach to inflammatory skin is directed at restoring the skin barrier function and reducing dehydration, for example, atopic dermatitis [51]. The dehydration effect caused by the vehicle may limit the therapeutic effect and may also induce a chronic inflammatory response [57]. Unlike other delivery vehicles, the ointment base affects the active compound’s bioavailability due to its occlusion effect on the stratum corneum, which then increases drug penetration/diffusion across the skin [59].

Emulsions including lotion, cream, nano-/micro-emulsion and liposomes are alternative approaches for the topical delivery of both hydrophilic and lipophilic compounds. The compositions of the selected phospholipid type, surface charge and phase transition temperatures have been shown to affect the topical delivery of liposomes [60]. Emulsifying agents and oil-in-water emulsions may also lead to dehydration of the horny layer of the epidermis and can cause damage to the barrier [60,61,62]. Our ongoing study has also demonstrated that generic desoximetasone cream containing too many emulsifying agents such as Span 60 and Tween 60 results in a poor therapeutic outcome in the imiquimod induced psoriasis-like animal model (Figure 2).

On the other hand, a penetration enhancer is one of the most common and useful strategies to overcome natural defects, such as achieving the molecular weight of over 500 Dalton of the selected compound, or to increase the skin deposition/permeation amounts on the target site [61,63]. Chemicals such as short-chain fatty acids and surfactants called skin penetration enhancers, percutaneous absorption promoters or accelerants produced a shift in the C-H_2_ asymmetric/symmetric stretching vibration of the stratum corneum lipid, resulting in increased wavenumbers and a disordered intercellular lipid structure between corneocytes in the stratum corneum. This resulted in alteration of the skin permeability barrier and increased the drug’s skin flux [62,64,65,66,67]. However, when the skin was under an inflamed state or experienced acute barrier disruption, the impaired barrier also showed a shift of C-H_2_ asymmetric/symmetric stretching vibration to higher wavenumbers [68,69]. As a result, the impaired inflammatory skin barrier treated by a drug containing penetration enhancers may lead to increased barrier damage severity and a greater inflammatory response. To avoid the damaging effect caused by the formulation excipients such as emulsifying agents, alcohols, and chemical enhancers, the addition of moisturizers/humectants, such as glycerol and urea, is recommended [70,71].

Due to high specificity and potency, proteins and peptides have become increasingly considered as therapeutic methods for serious and complex diseases such as cancer or autoimmune diseases. Because of their unique structure and unstable physical-chemical properties, proteins and peptides require more attention during manufacture/formulation and the administration process. The topical application of proteins and peptides is an attractive option and highly recognized by patients compared with conventional systemic injections [72]. Transdermal delivery of proteins and peptides could be more effective by using chemical penetration enhancers, transfersomes, nanoparticles delivery system, or cell-penetrating peptide modified nanocarriers [73,74,75,76,77]. Although most allergic contact dermatitis antigens are small-molecular-weight molecules or haptens, they become immunogenic after conjugation with proteins in the skin; larger-molecular-weight peptides or proteins may also induce the inflammatory response [78]. For example, protein contact dermatitis triggered, by high-molecular-weight proteins such as meat or fish, is one of the common skin diseases in food-service and health-care workers [79,80,81]. Therefore, selection of an appropriate dose of protein/peptides in the topical delivery system should be evaluated more carefully.

### 4.2. The Effect of Topical Pharmaceutical Formulation Compositions

Microemulsion and liposomes are the two most popular topical delivery systems. They are biocompatible and easy to be assembled with various oils, fatty acids, and surfactants. They also provide long-term stability and enhanced solubilization for both hydrophilic and lipophilic compounds. However, the composition of the formulation may still affect the skin barrier. Hathout et al. demonstrated that after application, the microemulsion breaks, diffusing all components separately and deeply into the skin. The individual components of the microemulsions, such as fatty acids and surfactants, may disrupt the stratum corneum structure. For instance, oleic acid, which works as a penetration enhancer, reduced the order of stratum corneum lipids, which then induced phase separation. The surfactant, such as Tween 20, then interacted with the stratum corneum lipids [65,82]. Furthermore, Oleic acid-propylene glycol-based formulations induced a greater skin lipid bilayers disruption than oleic acid-mineral oil-based formulations [64]. The compositions of the formulation may alter the skin integrity, increasing the permeability of drugs; however, such manipulations may lead to more serious barrier damage and a greater inflammatory response in inflammatory skin. Thus, reducing the amounts of skin lipid bilayer perturbation ingredients is recommended for inflammatory skin formulation. In contrast, ingredients that may improve barrier recovery/homeostasis, such as agonists of peroxisome proliferator-activated receptor-γ, should be considered [83,84,85]. Table 1 lists some of the recommended ingredients and their benefits for topical formulation based on the literature review and our experience.

### 4.3. Skin Absorption Routes and Skin Deposition

The transport of drugs through the skin may be carried out through three potential pathways: (1) the transepidermal route, which reaches across the continuous stratum corneum and can then be divided into (a) the intercellular route between the corneocytes and (b) the transcellular route through the corneocytes and interleaving lipids; (2) the trans-appendageal route, including sweat ducts, hair follicles, and associated sebaceous glands; (3) the micro-scale route, which stratum corneum is removed from by tape stripping, ablation, abrasion, or micro-needles, for instance. The compound may use more than one absorption/penetration route regarding the physicochemical properties of the selected compounds. For example, the transcellular route is suitable for transporting hydrophilic compounds, while the intercellular and trans-appendageal route is for lipophilic compounds [70,86,87].

Formulation may increase the epidermis or dermis deposition amounts of selected compounds due to the limit of nature physicochemical properties. For example, an oil-in-water microemulsion gel enhances one and a half folds of the epidermis deposition amounts of curcumin compared with the conventional delivery system in the mice skin model [88]. Through loading in the bicontinuous microemulsion, an over 800 Dolton lipophilic compound may transport through both transepidermal and trans-appendageal routes and then increase the skin deposition amount three-fold and enhance the therapeutic effect more than the ethanol solvent carrier in the psoriasis-like animal model treatment study [89]. Try et al. reported that small polymeric nanoparticles (below 100 nm) loaded fluorescein can easily penetrate and accumulate selectively in the deep epidermis and hair follicles, while larger polymeric nanoparticles (around 300 nm) and fluorescein solution remain at the epidermis surface in two atopic dermatitis animal models [90]. Body wash might disrupt the skin barrier integrity and then increase 4- to 5-fold cumulative absorption amounts of zinc pyrithione compared with a water-based carrier [86]. Although using the formulation may increase drug skin deposition amounts, the ultimate goal of suitable topical formulation development should provide a safe and effective therapeutic approach for the management inflammatory skin diseases.

### 4.4. Inflammatory Skin and Irritation Test

The skin irritation test is a safety study used to evaluate and select a suitable formulation for the next step in the development of topical pharmaceutical formulation. Normally, researchers select the dorsal skin of untreated normal healthy mice as the material. After at least three consecutive days of applying the selected formulation, researchers examine the skin color and histology results to clarify whether the selected formulation induced an irritation response [87,91,92]. However, the barrier function between untreated normal healthy skin and inflammatory skin is quite different. This means that a small quantity of irritants may not achieve the minion concentration required to induce the redness and inflammation response in normal skin. In contrast, a very small quantity of irritants may evoke severe trans-epidermal water loss, skin erythema, and an inflammation response in inflammatory skin. For example, Tupker reported that patients with a history of atopic dermatitis had a higher trans-epidermal water loss value after irritant exposure than the control group of healthy subjects. These subjects had increased sensitivity to irritants, which may be due to an impaired barrier function and/or a dry skin condition [93]. Thus, we proposed the “inflammatory skin irritation test model” in our previous study [89]. Figure 3 illustrates the procedure of the inflammatory skin irritation test. Here, mice dorsal skin was applied with imiquimod for six consecutive days to induce inflammation of the skin, followed by the application of only selected formulation bases for another five days. The formulations that induced/maintained the irritation/inflammation response, evidenced by scales, erythema, and dryness on the mice dorsal skin, as well as the histopathological results, which maintained the features of epidermal hyperplasia, should be excluded from the candidate formulation.

### 4.5. Therapeutic Adherence, Cosmetic Acceptability Formulation and Barrier Repair Therapy

Therapeutic adherence affects the effectiveness of treatment. Effective strategies to improve patient compliance are very important. Adherence is affected by a patient’s characteristics as well as disease- and treatment-related factors. On the other hand, satisfaction with therapy, cosmetic acceptability, and complexity of the treatment regime may also affect treatment compliance [94]. Therefore, cosmetic acceptability formulation design should be a considerable strategy for topical pharmaceutical purposes [95]. Several studies have reported that cosmetic moisturizer generally appears well-tolerated and suitable for topical use on sensitive skin [96,97,98]. Moisturizers are standard adjuvant therapy for anti-inflammatory skin disorders in dermatology. However, not all moisturizers are beneficial for the skin barrier. Some moisturizers or skin care products may induce barrier disruption in sensitive skin [99]. Thus, Elias proposed the use of physiologic lipid-based barrier repair therapy comprised of the key stratum corneum lipids, ceramides, cholesterol, and fatty acids at the appropriate molar ratio of 3:1:1 in sufficient quantities to restore the abnormality barrier function and reduce skin inflammation [100,101]. Our previous study also demonstrated that an emollient carrier formulation design could provide sufficient skin hydration for a low-grade inflammatory condition and enhance penetration of active ingredients into the skin [89].

## 5. Future Concept for Topical Formulation Development

Human keratinocytes express various transport-associated enzymes and detoxifying metabolic enzymes, for instance, cytochromes P450 (CYP) enzymes. CYP enzymes participate in the metabolism of various compounds such as drugs, toxicants, and carcinogens [102]. CYP3A4 and CYP4A5 belong to the CYP3 subfamily and jointly take part in about 30% of metabolism of drugs [103]. Reverse transcription-polymerase chain reactions reveal constitutive expression of CYP 1A1, 1B1, 2B6, 2E1, 3A4, and 3A5 in keratinocytes [104,105,106,107,108,109]. The administration of dexamethasone to murine skin, which results in the induction of CYP isozymes, implies that murine skin contains several inducible CYP subfamily enzymes capable of processing the metabolism of a wide range of xenobiotics and endogenous compounds [110]. Once applied to the skin, the active ingredients may be absorbed by the skin, metabolized, or may interact with other substances [111]. Our previous study demonstrated that an intraperitoneally administrated CYP3A inhibitor-based formulation can increase the therapeutic effect of CYP3A substrate drugs in the ischemic stroke rat model [112]. Therefore, inhibiting skin CYP-mediated metabolism may have the potential to alter the dermal-pharmacokinetics of the CYP substrate active compounds, therefore enhancing their bioavailability and therapeutic effects for future topical pharmaceutical formulation design concepts.

## 6. Conclusions

In addition to a sufficient dosage, to exert a therapeutic effect by penetrating the stratum corneum to the target area, the formulation of a treatment for inflammatory skin diseases should also consider how to reduce skin irritation and barrier disruption from the excipients of the formulation base. Furthermore, the application of a pharmaceutical-cosmetic acceptability formulation design to provide stratum corneum moderate moisturizing and barrier repair for ingredients, inhibit the skin CYP-mediated metabolism for the CYP substrate active compounds, and improve treatment adherence should be considered for inflammatory skin diseases treatment.

## Figures and Tables

**Figure 1 ijms-22-06078-f001:**
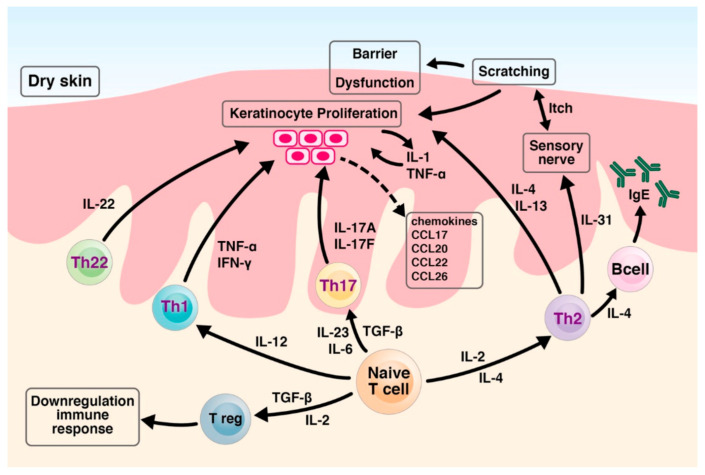
T cells involved in immune responses and the mechanisms of barrier dysfunction. Cytokines IL-12, IL-4, and transforming growth factor-β (TGF-β) regulate Th1, Th2, and Treg cell development, whilst IL-6, IL-23, and TGF-β promote Th17-cell differentiation, respectively. Unlike Th1, Th2, and Th17 cells, which mediate the skin inflammation process, Treg cells are involved in the down-regulation immune response. Th1, T helper 1 cell; Th2, T helper 2 cell; Th17, T helper 17 cell; Th2, T helper 22 cell; Treg, regulatory T cell.

**Figure 2 ijms-22-06078-f002:**
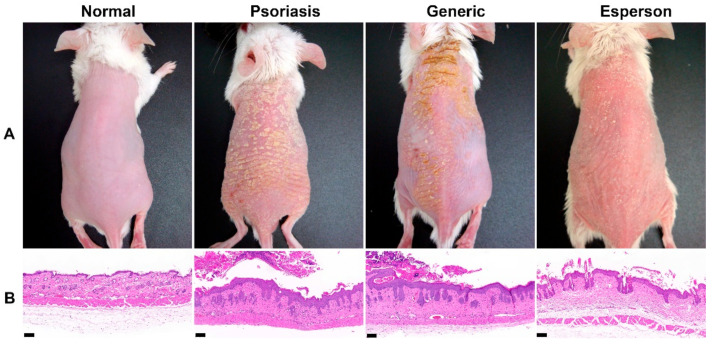
Study design of the inflammatory skin irritation test model: (**A**) morphology (**B**) histopathology by hematoxylin and eosin (H&E) staining. Scale bar = 100 μm. Normal, untreated healthy BALB/c mice; psoriasis, 5% imiquimod (Aldara, 3M Pharmaceuticals, Saint Paul, MN, USA) induced psoriasis-like mice; generic, 0.25% desoximetasone cream (Generic drug); esperson, 0.25% desoximetasone ointment (Sanofi, Handok Inc., Seoul, Korea).

**Figure 3 ijms-22-06078-f003:**
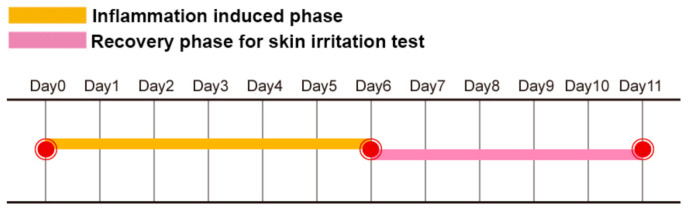
The study design of the inflammatory skin irritation test model.

**Table 1 ijms-22-06078-t001:** The recommended ingredients for topical formulation.

Ingredients	Benefits
Glycerol	Moisturizer
Linoleic acid	Agonists of peroxisome proliferator-activated receptor-γ, improved barrier recovery/homeostasis
Polyethylene glycol (PEG)-40 castor oil (RH-40)	Surfactant, increased viscosity and reduced the fluidity of the formulation
Silicon oil	Increased the occlusion effect on skin surface
Sorbitol	Moisturizer
Squalene	Component of human sebum, provided liquid lipid film on inflammatory skin surface which works as temporary barrier
Triglyceride	Component of human sebum, provided liquid lipid film on inflammatory skin surface, which works as temporary barrier

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
