# Peer review of "Strategies to Develop a Suitable Formulation for Inflammatory Skin Disease Treatment"

_ijms, 2021, doi:10.3390/ijms22116078_

Round 1

Reviewer 1 Report

The authors describe the review about topical delivery system and inflammatory skin barrier and provide opinions on strategies to develop a suitable formulation for inflammatory skin treatment.

The authors ought to add the description about some absorpted routs via the skin and the description about the difference of absorbed doses in the body part.

Author Response

Reviewer1

The authors describe the review about topical delivery system and inflammatory skin barrier and provide opinions on strategies to develop a suitable formulation for inflammatory skin treatment.

The authors ought to add the description about some absorpted routs via the skin and the description about the difference of absorbed doses in the body part.

Response: Thank you for your suggestions. We add the section 4.3. Skin absorption routs and skin deposition in the manuscript (page 9-10).

Reviewer 2 Report

While the title of this manuscript announces a very important area for a review, I felt its balance was unsatisfactory. While reading the title I was prepared to expect a real in-depth discussion on aspects of 'topical delivery'. However, at least the first half of the review is too superficial a commentary on a few inflammatory skin diseases in relation to the skin barrier. The second half focuses more on the main topical delivery aspects, but again much too superficially in my view.  

It fell short of an academic review that critically analyzes the subject in significant depth, and much of the content will appear in book chapters over recent years.

I was surprised that there was no mention of the skin microbiota, in this context., given its significant presence in/on the skin barrier. 

I found the inflammatory irritation skin test model, of significance, but felt that this was under-developed/explored in this review. 

I felt that potential caveats for inhibition of CYP-mediated metabolism in skin was not sufficiently explored as a issue.  

Minor:

There is some repetition, especially in the first half. 

Most melanoma (>70%) do not arise from pre-exciting lesions. 

There are several other cell types in the human epidermis, beyond keratinocytes (Langerhans cells, Merkel cells, melanocytes) that need to be discussed in this context.  

Author Response

Reviewer2

While the title of this manuscript announces a very important area for a review, I felt its balance was unsatisfactory. While reading the title I was prepared to expect a real in-depth discussion on aspects of 'topical delivery'. However, at least the first half of the review is too superficial a commentary on a few inflammatory skin diseases in relation to the skin barrier. The second half focuses more on the main topical delivery aspects, but again much too superficially in my view.  

It fell short of an academic review that critically analyzes the subject in significant depth, and much of the content will appear in book chapters over recent years.

Response: We appreciate your comments. This article briefly reviews the available data on the issues of topical treatment for impaired inflammatory skin barrier and provides opinions on strategies to develop a suitable formulation for inflammatory skin disease treatment. We change the title of the revised version as “Strategies to Develop a Suitable Formulation for Inflammatory Skin Diseases Treatment”.

I was surprised that there was no mention of the skin microbiota, in this context., given its significant presence in/on the skin barrier. 

Response: Thank you for your suggestions. We added a new section 3. Skin microbiota in the manuscript to discuss the relationships between skin microbiota and skin barrier (page 6- 7)

I found the inflammatory irritation skin test model, of significance, but felt that this was under-developed/explored in this review. 

Response: The inflammatory irritation skin test model was well developed and used in our formulation selection study for psoriasis treatment (Ref 92: Pharmaceutics 2020, 12, doi:10.3390/pharmaceutics12050457).

I felt that potential caveats for inhibition of CYP-mediated metabolism in skin was not sufficiently explored as a issue.  

Response: Our previous study demonstrated that an intraperitoneally administrated CYP3A inhibitor-based formulation can increase the therapeutic effect of substrate drugs of CYP3A in the ischemic stroke rat model. (Ref. 117: Pharmaceutics 2020, 12, doi:10.3390/pharmaceutics12080737). We believed that inhibition of CYP-mediated metabolism in the skin could be a promising concept for the formulation developing of CYP substrate drugs shortly.

Minor:

There is some repetition, especially in the first half. 

Response: Thank you for your suggestions. We revised the section 2.2.3 Psoriasis (page 4).

Most melanoma (>70%) do not arise from pre-exciting lesions. 

There are several other cell types in the human epidermis, beyond keratinocytes (Langerhans cells, Merkel cells, melanocytes) that need to be discussed in this context.  

Response: Thank you for your suggestions. We revised the section 2.2.4 Melamoma. (page 4-5)
